

# Constructed Technosols are key to the sustainable development of urban green infrastructure

Maha Deeb[1,6,9], Peter M. Groffman[1,6], Manuel Blouin[2], Sara Perl Egendorf[1,3], Alan Vergnes[4], Viacheslav

Vasenev[7,6], Donna L. Cao[3], Daniel Walsh[5], Tatiana Morin[6], Geoffroy Séré[8]

[1]Advanced Science Research Center at the Graduate Center of the City University of New York, NY 10031, USA
[2]Agroécologie, AgroSup Dijon, INRA, University of Bourgogne Franche-Comté, Dijon 21078, France
[3]Brooklyn College of The City University of New York, Department of Earth and Environmental Sciences, Brooklyn, NY 11210, USA
[4]Biologie - Ecologie - Environnement, Université Paul-Valéry Montpellier 3, Montpellier 34090, France
[5]Columbia University, Lamont Doherty Earth Observatory, NY 10964, USA
[6]New York City Urban Soils Institute, 2900 Bedford Avenue, Brooklyn, NY 11210, USA
[7]Landscape Design and Sustainable Ecosystems Department, RUDN University, Moscow117198, Russian Federation
[8] Laboratoire Sols et Environnement, Université de Lorraine, INRA, UMR 1120, F-54518 Vandœuvre-lès-Nancy, France
[9]Laboratoire Interdisciplinaire des Environnements Continentaux, Université de Lorraine, UMR 7360 CNRS, Metz, France

*Correspondence to*: Maha Deeb (mahadeeb.y@gmail.com)

**Abstract.** With the rise in urban population comes a demand for solutions to offset environmental problems caused by urbanization. Green infrastructure (GI) refers to engineered features that provide multi-ecological functions in urban spaces. Soils are a fundamental component of GI, playing key roles in supporting plant growth, infiltration, and biological activities that contribute to maintenance of air and water quality. However, urban soils are often physically, chemically or biologically unsuitable for use in GI features. Constructed Technosols (CT), consisting of mixtures of organic and mineral waste, are man-made soils designed to meet specific requirements and have great potential for use in GI. This review covers 1) current methods to create CT adapted for various GI designs and 2) published examples where CT have been used in GI. We address the main steps for building CT, the materials and which formulae that should be used to design functional CT, and the technical constraints to using CT for applications in parks, streetside trees, stormwater management, urban farming, and abandoned land. The analysis suggests that the composition and structure of CT should and can be adapted to available wastes and by-products and to future land use and environmental conditions. CT have a high potential to provide multiple soil functions in diverse situations and to contribute to greening efforts in cities (and beyond) across the world.





## 1 Introduction

### 1.1 Environmental issues in urban areas

Urban areas are exposed to multiple global (*e.g.* climate change, biodiversity loss) (Czech et al., 2000; McKinney, 2002; Shochat et al., 2006; Zalasiewicz et al., 2008) and local (*e.g.* urban heat island, increased air pollution, altered disturbance
regimes) (Bridgman et al., 1995; Lovett et al., 2000; Rao et al., 2014) scale environmental changes (Pickett et al., 2011). Urban development decreases forest and agricultural areas (Geneletti et al., 2017), which reduces ecosystem benefits (Foley et al., 2005) due to soil sealing and alteration of soil properties (Pistocchi et al., 2015; Scalenghe & Marsan, 2009). Urban development fragments landscapes and leads to a loss of connectivity between habitats (Haddad et al., 2015; Madadi et al., 2017; Vergnes et al., 2012). These changes have marked effects on multiple ecosystem services linked to air and water quality
(Cariolet et al., 2018; Doni et al., 2018; Jiang et al., 2018; Khan et al., 2018; Latif et al., 2018; Maurer et al., 2019), biodiversity (Guilland et al., 2018; McKinney, 2008; Pouyat et al., 2010) and human health and well-being (Das et al., 2018; Halonen et al., 2015; Säumel et al., 2012).

### 1.2 Green infrastructures: definition, benefits & challenges

Green infrastructure has emerged as an important approach to urban environmental issues over the past 20 years. Green
infrastructure is defined as engineered environmental design features built in interconnected natural and urban spaces to provide multiple ecological functions (Maes et al., 2015). Green infrastructure features can be completely human constructed, such as green roofs and green walls, or can be based on an existing ecosystem with some intervention, such as parks, urban farms, and bioswales. Ecological functions associated with green infrastructure include habitat for biodiversity (Vergnes et al., 2012), improving air and water quality (Pugh et al., 2012), reducing stormwater flows into sanitary sewer networks (Lucas &
Sample, 2015), filtering noise, microclimate stabilization (O'Neill et al., 2018), food supply (Specht et al., 2014), improving psychological and physiological human health (Tzoulas et al., 2007), and strengthening social relationships by attracting people to outdoor spaces (Coley et al., 1997). Economic advantages include increased tourism, reduced energy use for indoor temperature regulation, lower wastewater treatment costs, and increased property values and tax revenue (Mell et al., 2016).

Due to its ability to supply a wide range of ecosystem benefits (Flores et al., 1998; Kazemi et al., 2011; Keeley et al., 2013;
Lee & Maheswaran, 2011; Opdam et al., 2006; Sandström, 2002), green infrastructure has attracted significant attention in both developed (Brunner & Cozens, 2013; European Environment Agency, 2017; Davis Allen P. et al., 2009; Hall, 2010; Kabisch & Haase, 2013; Lewis, 2005; Pauleit et al., 2005) and developing (Akmar et al., 2011; Byomkesh et al., 2012; Jim, 2005; Qureshi et al., 2010; Rafiee et al., 2009) countries and is now an important component of city planning around the world (Tan et al., 2013; Zhao et al., 2013; Zhou & Wang, 2011). Interest in the use of green infrastructure is likely to increase along
with the ongoing expansion of cities.

There are numerous challenges associated with the development of green infrastructure (Haaland & van den Bosch, 2015; Jim, 1998). Building green infrastructure in densifying cities may pose challenges as there may be a conflict of interest among

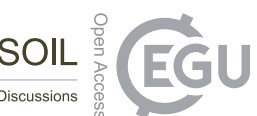

officials in regard to using the land for green spaces as opposed to residential, commercial, or service uses (Haaland & van den Bosch, 2015). Building a new green space requires a large of amount of financial and natural resources such as human labor, natural topsoils, and other imported materials. These costs are challenging because there is no easily measurable value of the economic benefits of green infrastructure (Haaland & van den Bosch, 2015).

Green infrastructures must provide suitable environmental conditions to foster the growth of vegetation. There is a need for sufficient sunlight and high-quality water for plants to grow, low foot traffic to avoid soil compaction, and an adequate surface area for tree growth. There is a potential for high mortality of plants due to suboptimal levels of water, light and nutrients, contamination, compaction, and other urban stressors.

## 1.3 Need for functional soils in urban areas

Soils in urban areas, especially non sealed ones, are highly heterogeneous, varying from natural (e.g., relict forest soils) to fully artificial (De Kimpe & Morel, 2000; Huot et al., 2017). Many urban soils are derived from a combination of exogenous, anthropogenic (so called technic materials) and natural geologic parent materials (Lehman, 2006). Urban soils have altered disturbance regimes from human activities (Godefroid & Koedam, 2007) and are frequently exposed to high volumes of stormwater surface runoff (McGrane, 2016). As a consequence, urban soils frequently exhibit poor biological, physical and chemical conditions that directly affect plant growth and reduce their ability to provide multiple ecological functions (De Kimpe & Morel, 2000; Morel et al., 2014).

Soils are a fundamental component of green infrastructure, playing key roles in supporting plant growth, infiltration of runoff and microbial activities relevant to nutrient cycling and pollutant degradation (Deeb et al., 2018; Keeley et al., 2013). To ensure the development of vegetation and offset the low fertility of urban soils, large quantities of topsoil are imported from surrounding agricultural or forest areas (e.g., 3 million of $m^3$ $year^{-1}$ in France) (Rokia et al., 2014). Taking soil from rural areas for urban use is not sustainable as it is costly and is accompanied by environmental risks, e.g., release of carbon dioxide (Walsh et al., 2018, 2019). Soil is a complex ecosystem and a limited resource that forms slowly, e.g., the rate of soil production has been estimated to be between 20-100 m $My^{-1}$ (i.e. 2-10 mm per century) (Heimsath et al., 1997). Thus, soil cannot be considered to be a "readily renewable resource that can be excavated and easily transported from rural to urban areas (Walsh et al., 2019). These concerns have stimulated interest in the development and use of constructed soils for multiple urban uses.

## 1.4 Existing solutions to provide functional soils in cities

Solutions to the challenges of greening cities depend on the degree and type of soil degradation, the affected area, and the size of the project. The most widely applied solution is the complete removal of the topsoil and replacement with natural arable soil imported from non-urban areas (Dick et al., 2006; Dickinson, 2000). As noted above, it is expensive to transport soil, and removal of the old soil creates a problematic waste product. This solution is particularly difficult to implement in developing countries (Bradshaw, 1997; Hüttl & Bradshaw, 2000).


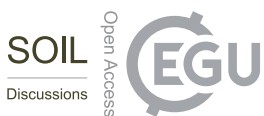

Another, less common solution to urban soil problems that has been applied in specific cases on smaller scales is using organic
amendments present in the city, such as composts, sludge, industrial by-products, and biosolids, to improve topsoil
characteristics (Larney & Angers, 2012). Adding organic wastes to restore ecosystem functions has received great attention in
different scientific studies (Basta et al., 2016; Gómez-Sagasti et al., 2018; Kumar et al., 1985; McGeehan, 2012). Moreover,
inexpensive sources of waste and by-products have become increasingly available in recent years, allowing for improvement
of soil quality while reducing landfill material (Vetterlein & Hüttl, 1999). The main drawback of this solution is concern about
the transfer of pollutants, especially Pb, from wastes to groundwater and runoff and subsequently to organisms (Battaglia et
al., 2007; Calace et al., 2005). Few studies have addressed the long-term sustainability of reclamation by organic waste
(Bacholle et al., 2006; Nemati et al., 2000; Vetterlein & Hüttl, 1999; Villar et al., 1997). Key concerns in this regard are that
composting is not widely practiced or integrated in policies worldwide and the amount of available organic waste varies widely
within and between cities (Narkhede et al., 2010).

Adding phosphate-bearing amendments such as chemical fertilizers or fish bone waste has been found to stabilize lead in urban
agriculture (Ruby et al., 1994). Phosphate amendments bond to unstable lead and form insoluble pyromorphite minerals (Ma
et al., 1995; Paltseva et al., 2018a, 2018b). The long-term stability of these minerals is uncertain, however, as they can change
with soil chemical conditions. Moreover, adding phosphate can mobilize arsenic through competitive anion exchange (Creger
& Peryea, 1994; Paltseva et al., 2018a). In addition, chemical fertilizers can have a variety of negative impacts on air and water
quality in the surrounding environment.

Using native instead of non-native plants has been proposed as a reclamation solution in urban landscapes (Ries et al., 2001)
for stormwater management (Bartens et al., 2008; Culbertson & Hutchinson, 2004; Lucas & Greenway, 2007; Selbig &
Balster, 2010), land use sustainability (Dornbush, 2004; O'Dell et al., 2007), conservation of wildlife habitat (Fletcher &
Koford, 2002), erosion control (Beyers, 2004), carbon sequestration (Isaacs et al., 2009), and soil remediation (Swedo et al.,
2008) and stabilization. Planting native species has an indirect influence on soils by promoting biodiversity and creating
resilient communities. However, there is no guarantee that native plants will grow in soils with significant anthropogenic
impacts (Suding et al., 2004). Moreover, this solution has been practiced mostly to restore natural landscapes and to reverse
species loss (Richardson et al., 2007).

## 1.5 Opportunities to construct new soils

The "new" idea of building constructed Technosols is based on combining biotic and abiotic characteristics of diverse materials
to produce specific functions and benefits (Fig. 1). Technosols are defined as deliberate mixtures of organic and mineral wastes
and by-products constructed to meet specific requirements (Craul, 1999; Damas & Coulon, 2016; Rokia et al., 2014; Séré et
al., 2008; Yilmaz et al., 2018). The process of constructing Technosols includes existing approaches such as recycling organic
materials as well as natural and anthropogenic parent materials, and including native plants. The use of easily available
materials provides economic benefits (Walsh et al., 2019), has limited impact on natural resources, and makes use of waste
materials. Constructed Technosols can be created as new buildings are constructed from excavation and demolition waste,

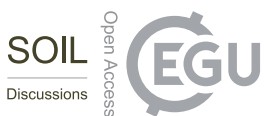

saving on labor and transportation costs for both waste disposal and importation of soils. Because the overall cost of constructed soils is lower, green spaces built with constructed Technosols can more easily be integrated with city planning. These advantages can be particularly important in cities or neighborhoods with fiscal constraints, thus alleviating some environmental

justice issues.

In the sections below, we review: 1) current methods to create CT adapted for various green infrastructure designs and 2) published examples where CT have been used in green infrastructure.

## 2 Materials & Methods

### 2.1 What is a constructed Technosol?

Anthropization is the process of soil formation in urban environments. The World Reference Base for Soil Resources (WRB, 2014) defines Technosols as soils which contain greater than 20% artificial materials by volume and are within the first 100 cm of the land surface. At the moment there is no universal definition of constructed Technosols. Séré et al. (2008) defined the construction of a Technosol as a process "using wastes and industrial by-products which are formulated and stacked in layers to build a new soil profile over *in situ* degraded substrates". Prokofyeva et al. (2011), on the other hand, defines

constructed Technosols ("constructozems") as bodies of soil with complicated stratification greater than 40–50 cm in thickness and created for special purposes such as salt protection in semi and severely dry areas (Smagin, 2012). In this review, we will define constructed Technosols as the result of the voluntary action of creating a "soil" (soil made by humans) using artefacts (*i.e.* technogenic materials, particular wastes, or semi-natural materials including deep materials such as sediments or soil material from C horizons), and intentionally shaping them to provide a suitable environment for vegetation growth. The choice

of the vegetation may be for aesthetic (greenery), protective (wind erosion and stormwater) or productive (agriculture) roles (Fig. 1). This new soil could be created for use in multiple green infrastructure designs including squares and parks, accompaniments for public buildings (tree lined streets, flower beds, verges, pocket spaces, green roofs), accompaniments for traffic lanes (roads and railway lines), stormwater management, urban farming, household yards, and abandoned land reclamation (Fig. 2).

### 2.2 Bibliographic search

Structured and semi-structured searches were conducted using major scientific databases (Scopus, Web of Knowledge and Google Scholar) along with the appropriate cross-referencing to obtain supporting literature. Additional searches were based on the authors' own knowledge of potentially relevant work and their experience with constructed Technosols. Structured searches were carried out with the following keywords (title, abstract, keywords): Constructed Technosols, engineered soils,

Anthroposols, green space, parks, green infrastructure, green roof, stormwater management, street trees, ecosystem services, densification, abandoned land, food security, soil functions, biodiversity, urban farming, recycling, landfill, and waste management.



The literature considered for the review was published in English, French, and Russian. Other literatures that were difficult to translate or locate were excluded from this review.

## 3 Results and discussion

### 3.1 Steps for the construction of a Technosol

### 3.1.1 Adapting the characteristics of constructed Technosols to the requirements of green infrastructure

Green infrastructure features require functional soils or adapted substrates in order to be fully operational. Key characteristics that should be considered for the construction of Technosols include:

- Adapted chemical fertility (Rokia et al., 2014) suitable for specific vegetation with specific features (*e.g.* trees, lawns, grasslands, flower beds);
- Soil depth sufficient for vegetation anchorage (Marié X & Rossignol J.P, 1997), *e.g.,* trees with deep roots and high root system density;
- Concentrations of contaminants must be compatible with the actual use of public spaces (Egendorf et al., 2018) in terms of health risks (*e.g.* inhalation, soil ingestion) and land use (*e.g.* urban agriculture);
- Bearing capacity must be compatible with trampling, parking or vehicle traffic (Grabosky & Basset, 1998);
- Water infiltration is often desired to limit floods (Liu et al., 2014), which requires permeable soils;
- Water storage capacity to support green lawns and ornamental plants with limited irrigation (Smagin and Sadovnikova, 2015; Vasenev et al., 2017);
- Low bulk density and shallow root system depth (Vijayaraghavan, 2016) are required for specific substrates that are used on buildings (*e.g.* green roof);
- Minimal long-term maintenance *e.g.*, the addition of organic matter and frequent watering is not necessary;
- Soil structure stability is often desired to limit erosion and dust respiration (Deeb et al., 2017);
- High hydraulic conductivity is required for land uses such as green roofs and stormwater management systems;
- Moderate ratios of mineral and organic content, generally refraining from the addition of more than 30% organic matter is desired (Deeb, 2016a; Grosbellet, 2008; Vijayaraghavan, 2016) to avoid excessive loss of organic matter by oxidation ($CO_2$ emissions) or leaching, which will change soil volume.

These characteristics vary depending on the intended uses and functions of the green infrastructure in question (Fig. 2, Table 1). For example: i) tree-lined streets should have stronger physical support functions than green roofs; ii) food production in community gardens requires parent materials with very low contamination; iii) organic contaminant decomposition and water infiltration are important for bioswales (green infrastructure for stormwater management).



### 3.1.2 Use of waste materials in constructed Technosols

The generation of waste is a negative anthropogenic impact, especially in urban areas, which are estimated to produce 2.2 billion tons of solid waste per year by 2025 with a cost of $375.5 billion (Wilson & Velis, 2015). Efforts to minimize solid waste disposal in landfills and to encourage recycling are increasingly common (Krook et al., 2012). Constructed Technosols are a viable solution for waste management.

There is a huge diversity of wastes and by-products generated in cities. However, not all of them are considered relevant for the construction of Technosols (Table. 2). Some researchers (Deeb et al., 2016a, b, 2017; Pruvost et al., 2018; Walsh et al., 2018, 2019) have focused on construction and demolition waste (C&D) mixed with organic waste because C&D debris generate high percentages (around 35%) of the total solid waste worldwide (Hendriks & Pietersen, 2000). In Europe, C&D debris comprise 46% of total waste (Eurostat, 2017). Walsh et al. (2019) showed that an estimated $1.7 \times 10^6$ tons of unused, uncontaminated native soil, consisting mainly of glacial sediments, is generated each year from building construction in NYC alone. Disposal of this clean material requires 60,000 truck transports and $8.7 \times 10^6$ km of travel distance, using $4.3 \times 10^6$ L of petroleum products, emitting 11,800 tons of $CO_2$, and costing over $60 million USD per year. Despite the large volumes of C&D waste produced, only small fractions are generally recycled (Gálvez-Martos et al., 2018).

The SITERRE project (Damas & Coulon, 2016), one of the first and largest projects to create constructed Technosols for greening urban spaces, was based on the European catalog of 836 urban waste types. Three specific criteria were used to choose wastes for use in constructed Technsols:

1. Non-toxic: the waste should not be classified as dangerous for human health or living organisms.
2. Easy to handle: liquid and pasty wastes were not considered and only granular solid materials could be used.
3. Appearance: the selected wastes have to generate minimal disturbances to the local population and also be compatible with the lifestyle in densely populated urban areas, *i.e.* organoleptic criteria such as smell or color.

After satisfying the fundamental criteria above, the selected waste should additionally comply with at least one of the following

1. Fertile: the materials should contribute positively to the germination or development of vegetation.
2. Bearing capacity: the materials should have a positive impact in terms of geotechnical properties, *i.e.* soil stability, plasticity and cohesion.

Based on these five criteria, five mineral and six organic by-products were chosen (Damas & Coulon, 2016) including: i) materials from excavated acidic soil deep horizons, ii) materials from excavated basic soil deep horizons, iii) bricks, iv) concrete, iv) unsorted demolition rubble, v) track ballast (the material upon which railroad ties are laid), vi) green wastes, vii) sewage sludge, viii) green wastes compost, ix) sewage sludge compost, x) paper-mill sludge and xi) street sweeping wastes. Other researchers have investigated different waste materials including coffee grounds (Grard et al., 2015), backfill waste (Vergnes et al., 2017), thermally treated industrial soil (Séré et al., 2010), recycled ferrihydrites (Flores-Ramírez et al., 2018), and glass (NYCGIP, 2011).





### 3.1.3 Construction of functional Technosols

A specific formula/mixture of waste ingredients must be determined to achieve expected functions. Twenty-five combinations were made by Rokia et al, (2014) using the wastes selected in the SITERRE project, either binary mixtures at 5 different ratios (0/100, 20/80, 50/50, 80/20, and 100/0, volume/volume), or ternary mixtures with 60% v/v coarse mineral material and 40% v/v mixtures of organic and mineral mixtures. While no single stream waste material was adequate as soil or horticultural substrate in isolation, the mixtures produced a range of chemical and physical properties. Both brick waste and excavated deep

horizons mixed with compost produced physically and chemically fertile substrates, based on the chemical and physical characteristics of fertile soils, but not on their ability to sustain plant growth (Rokia et al., 2014). Mathematical models were developed to simulate the characteristics of mixtures from the characteristics of single materials (Rokia et al., 2014).

### 3.1.4 Technical constraints to consider while constructing Technosols

There are constraints that need to be considered when planning to use constructed Technosols for developing green
infrastructure. Some waste materials are extremely heterogeneous and thus difficult to characterize to ensure their safety. In addition, different materials offer a wide range of physicochemical properties. For example, even though brick and organic matter showed optimal chemical characteristics for plant growth in the SITERRE project (Rokia et al., 2014), these materials showed low potential for aggregation compared to excavated deep horizon material combined with organic matter (Vidal-Beaudet et al., 2016). Fourvel et al. (2018) compared six types of degraded excavated sediments with heterogeneous
characteristics from France and found that fertility varied greatly among the sediment types.

Mixed waste materials need a specific time period to form a stable structure, thus erosion, runoff, and compaction are risks in the early stages of soil formation. Establishment of plants with high density root systems (Deeb et al., 2017) or building erosion control barriers around the constructed Technosols could reduce these risks.

Fresh organic waste can be problematic as it can have a toxic effect on plant growth (Yilmaz et al., 2016), sometimes by
creating anoxic conditions. Thoughtful choices, e.g., using mature compost or mixing in mineral material that drains well (such as sand) can avoid anoxic conditions.

Finally, one of the most important limits is social rejection, especially since some organic waste (e.g., sewage) can have an objectionable odor. This problem can be avoided by applying such materials in deeper horizons. However, finding a solution for the general unpopularity of using waste for developing green spaces could be more challenging as it requires specific
education, observation over time, cooperation between biophysical and social science disciplines, and active engagement with communities. It can be expected that the increasing re-use of materials considered to be waste will influence norms and policies regulating their production, in a way which could ease their recycling in the future.



### 3.1.5 Pedogenesis in constructed Technosols

The process of pedogenesis (soil formation), which is strongly controlled by local environmental conditions dictates that the
age of Technosols is a key driver of their properties (Hui et al., 2017). Séré et al. (2010) demonstrated that artificial soils
underwent pedogenic processes similar to natural soils (e.g. chemical weathering, soil structure evolution, horizonation) and
Deeb et al. (2016a) showed that after one cycle of humidification (one week, in laboratory conditions), constructed Technosols
were able to develop hydrostructural behaviors similar to natural soils. After five months, the same Technosols showed a high
ability to aggregate, and in some cases enrich the carbon content of particular size fraction of aggregates (Deeb et al., 2017).
High mineralization of organic matter may occur in the early stages of the pedogenesis of constructed Technosols, especially
when a high quantity of organic matter is used (Grosbellet, 2008). Because the loss of organic matter will vary with
environmental conditions, modifying the design of constructed Technosol could be a solution. For example, in arid climates
where the risk of hydromorphy is low, one can add a higher concentration of organic matter in subsurface deep horizons than
in surface horizons, so that organic matter will be protected from oxidation and will improve water availability surrounding
root system. Establishing an active plant community as an internal source of organic matter and as a driver of multiple biotic
and abiotic interactions is also important (Deeb et al., 2016b; Jangorzo et al., 2018, 2014; Pey et al., 2013a, 2014). A three-
year monitoring plan of carbon balance in Technosols from Moscow, Russia showed substantial carbon release during the first
months after construction, which was almost completely fixated in the next three years by carbon input from the root biomass
growth (Shchepeleva et al., 2017; 2019). Given the specific growth requirements of different species, plant selection must be
done with great care (Pruvost et al., 2018).

## 3.2 Adaptation of constructed Technosol to specific contexts

### 3.2.1 Constructed Technosols for parks and squares with lawns

The majority of studies of implementing parks or squares attempt to mimic natural soils by constructing Technosols with one
or two distinct horizons. Horizon A (growing horizon) is generally designed to support germination and initial growth of
grasses; as a result, high amounts of organic wastes are often used as dominant ingredients. Horizon B (technical horizon) is
commonly designed to provide high stormwater retention and to prevent the leaching of organic carbon and nitrate (Fig. 3a).
Deeb et al. (2016 a, b, c, 2017) defined optimal waste mixtures to fulfill specific soil functions such as erosion control, carbon
and water storage, structural stability, and biomass production for the growing horizon. These authors built constructed
Technosols under specific climate conditions and mixed different amount of compost (0%, 10%, 20%, 30%, 40% and 50%
v/v) with demolition waste to optimize production of a grass, *Lolium perenne*. Depending on the desired function of soils,
different ratios of compost will complement certain soil functions more than others. For example, biomass production generally
increases with increasing compost ratios. Structural stability was also found to increase with compost ratio, but only in the
presence of plants and macrofauna, the main drivers of soil structure. In terms of hydro-structural properties, the mixtures
ranging from 20 to 40% compost exhibited similar characteristics. Therefore, it may be unnecessary to use excessive amounts



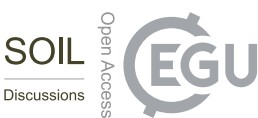

280 of compost to achieve optimal available water content in macro and micropores. The interaction between parent materials (compost ratio) and biota played a major role in water storage and available water, explaining 39 % of the observed variance of these functions.

Other organic parent materials, such as uncomposted green waste (tree & grass cuttings) or urban sewage sludge, have also been evaluated. Yilmaz et al. (2018) studied four constructed Technosols for the development of parks, squares, lawns, and

285 trees in *in situ* lysimeters. The artificial soils were either planted with trees (*Acer platanoides*) or ryegrass (*Lolium perenne L.*). The authors studied specific soil functions such as soil structure arrangement and water maintenance through analyzing hydraulic conductivity, macro-micro porosity, organic matter content, and water availability. The results demonstrated that compared to natural soils, constructed Technosols provided high porosity, abundant water storage for plant use, and high production of trees and ryegrass likely due to the high organic matter content and soil pH of the Technosols. The study

290 concluded that constructed Technosols have the capabilities to support vegetation growth for urban greening.

Our analysis of results from previous studies supports the recommendation of using a soil mixture with 20 to 30% compost for parks and lawns to avoid mineralization and nutrient leaching. In addition, because macrofauna and macroflora have a positive effect on structural stability, porosity, and carbon storage, it is recommended to integrate biota, e.g. earthworms, as soon as possible when planning to build parks with constructed Technosols (Deeb et al., 2016c, 2017). Several ways exist to integrate

295 biota: one should conserve macrofauna in the rare cases where it is already present in materials. When absent, one could also favor its recolonization from populated areas by planning ecological corridors or by inoculation with soil blocks or ex situ bred populations (Blouin et al., 2013). Vergnes et al., (2017) surveyed habitat colonization of parks built using a mixture of excavated deep horizons and natural topsoils and noted that parks that had an addition of topsoil had significantly more earthworms and ants compared to parks without topsoil addition.

300 Pruvost et al. (2018) in a four-year field experiment, tested the potential of excavated soil deep horizons alone or mixed with green waste compost and/or concrete waste, to support the growth of six tree species for the development of parks. To assess the fertility of the recycled substrates, they measured tree growth, soil physicochemical characteristics, and soil macrofauna. Excavated deep horizons, independently and with other mixtures, showed promising results. The mixture with the highest tree mortality was excavated deep horizon and green waste compost. However, the addition of crushed concrete to this mixture,

305 counteracted the negative effects and decreased tree mortality. The mixture with 10% green waste compost and 70% crushed concrete had the most favorable outcome with the highest tree survival, growth rate, and fastest soil macrofauna colonization. *Acer campestre* and *Prunus avium* were the tree species in the parks with no mortality and are likely the most adaptable. These results highlight the importance of balancing both the soil mixture and species choice (Fig. 1).

### 3.2.2 Constructed Technosols for developing tree lined streets

310 Urban street trees frequently exhibit high mortality due to multiple factors (Jim, 1998; Rossignol, 1999) including limited nutrients (Li et al., 2013), contamination (Muir & McCune, 1988), poor soil physical properties (Lindsey & Bassuk, 1992), inadequate light (Jim, 1998) and insufficient space for roots (Dubik et al., 1990; Lindsey & Bassuk, 1992). Additional literature



has documented a conflict between street trees and urban infrastructure (sidewalks, sewers, electricity cables, etc.) (Lindsey & Bassuk, 1992). To avoid these conflicts, recommendations have been developed for suitable tree species (McPherson et al.,
2016), their spatial distribution (Thomsen et al., 2016) and for providing sufficient surface area for root systems. Moreover, several researchers have documented how engineered (constructed) soils can reduce tree morality and conflicts with infrastructure (Rossignol, 1999). The main idea of these recommendations is to limit soil compaction and maximize water storage capacity and to avoid tree species that have a short life cycle and that damage surrounding infrastructure. For example, Daunay (1999) used horizons of different mixtures of natural and artificial materials including a 100 – 150 cm layer of 65%
ground-stone and 35% soil mixture, overlain by 30 – 50 cm of organic material. The ground-stone mixture provided a firm foundation for plants with low risk of compaction, creating a resilient soil structure with high infiltration capacity and no inhibition of root growth.

Damas & Coulon (2016) developed and tested a model soil for street trees (Fig. 3b) that includes a base layer skeleton horizon composed of a mixture of coarse mineral parent materials (i.e., track ballast, concrete waste, demolition rubble) with low
content of organic materials, topped with a growing horizon layer of organic-rich materials for root growth. Cannavo et al. (2018) adopted a similar horizon order, using a layer of sand (0.15 m) as a base foundation for adequate drainage. For the skeleton horizon (1.85 m), they used three different mixtures; one contained fine mineral material, demolition waste, and green waste, the second consisted of fine mineral material track ballast, and sewage sludge, and the third used chalcedony and leaf litter. These were covered by a growing horizon (0.8 m) comprised of 60% crushed brick waste and 40% sewage sludge and
green waste. After three-years, the two skeleton horizon mixtures that contained fine mineral, demolition waste, track ballast, green waste, and sewage sludge showed greater plant development than the mixture with chalcedony and leaf litter alone (Cannavo et al., 2018) suggesting that the skeleton horizons are an important source of nutrients for plant growth. Other studies have documented the importance of the growing horizon, specifically the richness of the organic waste (Vidal-Beaudet et al., 2015) for tree rhizosphere development. While these constructed Technosols showed encouraging results for the growth and
development of street line trees, it is important to note that these results were the outcome of innovative design, where each detail such as the waste mixture, order of layers, depth, tree species, and etc., was meticulously considered as the soil was engineered.

### 3.2.3 Constructed Technosols for stormwater management

The soils in stormwater management systems face similar, if not greater anthropogenic stress than other urban soils.
Stormwater management systems are designed to filter stormwater that contains contaminants, with small areas receiving runoff from much larger areas of impervious surfaces, creating physical, chemical and biological stress for soils and plants. The main objective of stormwater management systems is to absorb runoff and avoid floods by improving water entry into the soil profile and by enhancing water storage within the soil, which increases water retention for vegetation use. A range of biogeochemical processes affect greenhouse gas emissions, organic carbon storage and the biofiltering of heavy metals and
organic contaminants in these features (Deeb et al., 2018; McPhillips & Walter, 2015; McPhillips Lauren et al., 2018).





Given the high stress conditions of stormwater management systems, there is great potential for use of constructed Technosols in these features to optimize water storage and infiltration by providing high porosity, permeable surface and supporting microbial activity by planting vegetation with root systems that are resilient against physicochemical pressures. Constructed Technosols have been used in several green infrastructure designs for stormwater management built by the Department of
Environmental Protection in New York City (NYDEP, 2011) including enhanced tree pits (ETP) and streetside infiltration swales (SSIS). Both ETP and SSIS are rectangular-shaped bioswales built on sidewalks adjacent to the street. ETPs typically cover an area of 9 m$^2$, with a 0.6 m constructed soil layer underlain by different depths of gravel, recycled glass, or storage chambers (Fig. 3c). SSIS have an average area of 19 m$^2$ but do not have layers of gravel, recycled glass, or storage chambers. Water from the street enters the ETP and SSIS through curb cuts and inlets into the stormwater management system (NYCGIP,
2011). The design (surface area, order of layers, distance to soil from street, and shape) of the stormwater management system affects bacterial community and function (Joyner et al., 2019). Significant levels of denitrification (a water quality maintenance process that converts nitrate, an important agent of eutrophication, into nitrogen gas) have been observed in these green infrastructure systems (Deeb et al., 2018) and at other sites as well (Bettez & Groffman, 2012; Morse et al., 2017). Other studies have found high levels of microbial diversity and activity (Gill et al., 2017) and improved infiltration rates in these
systems relative to urban soils (Alizadehtazi et al., 2016). The constructed stormwater management soils studied by Deeb et al. (2018) had low levels of contamination by metals and total petroleum hydrocarbons. These results suggest that there is a high potential for the use of constructed Technosols in stormwater systems although further research, in a wider variety of settings (e.g., higher contaminant levels) is needed.

### 3.2.4 Constructed Technosols for urban farming

There is a significant body of work examining the possibilities and challenges of urban farming, but most research on this topic does not necessarily examine the soil that literally lays the ground for such endeavors. The most common attention given to urban agriculture soils pertains to the presence of inorganic and organic contaminants (Brown et al., 2016; Kessler, 2013; Marquez-Bravo et al., 2016; McBride et al., 2014; Mitchell et al., 2014; Sipter et al., 2008; Spliethoff et al., 2016). Both ongoing and historic activities related to industrial processes, use of leaded paint and gasoline, and incineration have left a
legacy of potentially toxic elements and compounds in urban soils that create risks for gardeners and others who come into contact with soil (Chillrud et al., 1999; Laidlaw et al., 2017; Mielke et al., 1983; Root, 2015). While there are cases where urban farmers use soils formed from naturally deposited parent materials, most proactive urban farmers deliberately avoid such practices in order to mitigate contaminant exposure and enhance fertility. Most urban farmers grow plants in a wide range of constructed Technosols that are difficult to classify (i.e., raised beds built by composting local organic waste mixed with
potting soil). These are valuable safety practices, given research showing that community growing spaces are less contaminated than home gardens or yards (Cheng et al., 2015). These findings may be due to a variety of factors, including the use of imported soil materials and amendments that dilute or immobilize pollutants.



Researchers have examined the potential for constructed Technosols to support nonedible biomass (Rodrigues et al., 2019). Rokia et al. (2014) characterized the agronomic physical and chemical properties of a range of constructed Technosols built

by using a combination of waste materials. However, these authors did not evaluate biological properties, especially yield. Very few studies have performed field scale trials and analysis of constructed Technosols in urban agriculture settings. Brandon & Price (2007) investigated the use of sediments dredged from rivers and lakes and outlined an approach for manufacturing soil. Such dredged sediments have been found to be appropriate for agricultural purposes (Darmody & Marlin, 2002). Similarly, Egendorf et al. (2018) used excavated deep horizon sediments from the New York City Clean Soil Bank / PUREsoil

NYC program (Walsh et al., 2019) mixed with different percentages of compost (20, 33, and 50%) to create constructed Technosols. These constructed Technosols were used for field scale trials in urban farm settings and were found to be successful for urban agriculture (Egendorf et al., 2018). In these studies, the constructed Technosols exhibited low levels of contamination, which did not increase over the span of one year of exposure to the urban atmosphere. The results from these authors suggest that there is high potential for constructed Technosols to be used in urban agriculture.

Grard et al. (2018) highlighted the importance of constructed Technosol design, (e.g., the order of substrate layers to specific biological activities, Fig. 4a) in green roofs. This author studied multiple plots filled with various urban wastes such as green waste compost, shredded wood, crushed tiles and bricks, used coffee grounds, and a biowaste compost. The different Technosols were evaluated in terms of food production, fertility, and water retention. Results showed that the constructed Technosols exhibited low contamination levels, were fertile, and could sustain high quality food production for up to five

years.

As mentioned above, urban farmers and growers are often constructing their own Technosols when they import a variety of substrates as growing media. Similar to natural soils, applying crop rotation to constructed Technosols may improve fertility, stability, and crop production. In addition, choosing a variety of species with different root systems may improve soil stability and nutrient absorption. Community-based participatory research projects could help in collecting data about the effects of

amendments and other urban farmer practices on ecosystem services.

### 3.2.5 Constructed Technosols as a solution to reclaim derelict land

A major focus of urban sustainability programs has been brownfield sites and other types of degraded land. By definition, a brownfield site is a contaminated or degraded plot of land that formerly served as an industrial or commercial facility, but is no longer in use or operational (US EPA, 1997). Revitalizing brownfield sites, which are commonly located in low income

neighborhoods, may not only beautify these neighborhoods, but may also bring about economic opportunities that arise from developing commercial, industrial, or residential properties in place of the brownfield. Aside from economic opportunities, brownfield sites can also be recycled into green space or green infrastructure that will potentially introduce a wide variety of social, environmental, and health benefits. Although greening brownfields can potentially produce numerous benefits, they are challenging to develop due to safety and liability concerns as well as high planning, construction and maintenance costs (De

Sousa, 2003).





Soils are fundamental to brownfield redevelopment and there is a high potential for use of constructed Technosols in brownfield projects. For example, Séré et al. (2008) highlighted the importance of shaping a functional soil to restore abandoned land, focusing on three main soil functions: water buffering and transformation, biomass production, and trace element cycles. These authors noted additional functions such as microbial activity related to the nitrogen cycle (Hafeez et al.,

2012a, 2012b), improving soil structure (Jangorzo et al., 2018, 2013, 2014) and macrofauna activity related to nutrient cycling (Pey et al., 2013a, 2014). The French Scientific Interest Group (GISFI) (http://www.gisfi.prd.fr) conducted the first large-scale (100 m$^3$) field experiment using waste comparing two types of constructed Technosols for ecological reclamation of abandoned land (Fig. 4b) based on varying compositions and depths of three different materials: green waste compost, paper by-products, and treated industrial sediment wastes (Séré et al., 2008). These constructed Technosols provided several soil functions such

as water filtration to remove contaminants and nutrient supply for surrounding plants (Séré et al., 2008). Over time, they developed functional organo-mineral soil pedons (Séré et al., 2008) with active pedogenic process (Séré et al., 2010) and a wide range of biotic activities (Hafeez et al., 2012a; Pey et al., 2013b). These authors suggest that constructed Technosols have an important role to play in brownfield reclamation, but they make several recommendations that should be considered such as the choice of waste, especially in the early stages (Séré, 2007; Séré et al., 2008), the importance of stimulating pedogenic

process (Séré et al., 2008), and careful consideration of the targeted land use functions (Séré et al., 2008). Slukovskaya et al. (2019) demonstrated the advantages of Technosol construction to reclaim heavily contaminated lands in the Russian subarctic zone. Implementation of carbonatite and serpentinite-magnesite wastes covered by hydroponic vermiculite not only allowed to immobilize considerable amounts of Ni and Cu, but also increased biomass production, carbon sequestration, and microbiological activity.

**4 Conclusion**

This review provides evidence that the construction of Technosols for the design of urban green infrastructure is a valuable alternative solution to the consumption of natural resources such as soil materials, wood chips or peat coming from surrounding rural areas. Constructed Technosols can contribute to sustainable environments in urban contexts as they supply multiple functions and services in several land uses. Advantages to using Technosols include reduced economic costs associated with

deposition of materials in landfill sites, lower remedial costs, reduced spending on fertilizer; lower public health risks due to healthier air quality from decreased transportation of dusty waste and urban degraded soils, reduced soil and water pollution, increased vegetation which provides a cooler urban microclimate, lower greenhouse gas emissions, and greater food production in urban areas. However, this study confirmed that each element used in the formula to design constructed Technosols should be carefully considered. These elements include the ratio and the composition of waste, the order of horizons, the choice of

plant species, the implementation methods, and the critical need to foster pedogenic processes, especially during the first months following construction. There is a strong need for further research on how constructed Technosols can be used for





multiple purposes in sites across the world. This research will contribute to urban sustainability as well as to fundamental knowledge on soil formation and function processes.

**Author contribution**

Research goals and aims of this article are based on an original idea from Maha Deeb, and Geoffroy Séré helped carrying them out. Maha Deeb prepared the manuscript with contributions from all co-authors. Peter M. Groffman supervised the first author and amended the whole text with numerous improvements. Manuel Blouin, Sara Perl Egendorf, Alan Vergnes, Viacheslav Vasenev, Donna L. Cao, Daniel Walsh, Tatiana Morin all participated in editing based on their respective expertise.

**Competing interests**

The authors declare that they have no conflict of interest.

**Acknowledgement**

The research was supported by RUDN University's "5-100" project. Data from Russian case studies was collected and analyzed with the support of the Russian Science Foundation under project #19-77-300-12. The authors are thankful to Landry Collet for allowing the use of his pictures for publication (All photos from him unless stated otherwise).

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



**Table 1: Key parameters of constructed Technosol functions provided by different land use based on literature.**

| Land use | Soil functions | Key parameters | References |
|---|---|---|---|
| Parks and squares lawn | Erosion control | Macro-micro porosity, structure stability, infiltration, water storage, hydro-structural characteristics, water retention and shrinkages curves, particle size density, texture | Deeb et. al., (2016a, 2016b, 2017) |
| | Soil structure arrangement and water maintenance | Hydraulic conductivity, macro-micro porosity, organic matter, and available water content, aggregation, carbon/ nitrogen | Yilmaz et al., (2018); Deeb et al., (2017) |
| | Biomass production | Root/vegetation biomass, tree growth | Yilmaz et al., (2018); Deeb et al., (2017); Pruvost et al., (2018) |
| | Biological diversity | Abundance of earthworms and ants, soil macrofauna colonization | Pruvost et al., (2018); Vergnes et al., (2017) |
| | Microbial diversity and activity | Microbial biomass carbon, microbial respiration, microbial metabolic quotient | Vasenev (2011); Vasenev et al., (2017) |
| Tree lined streets | Soil structure arrangement | Total and macro porosity, bulk density | Cannavo et al., (2018) |
| | Nutrient cycle | Available phosphorus and potassium, carbon-nitrogen ratio, macro-micro nutrients, pH, cation exchange capacity | Cannavo et al., (2018) |
| | Biomass production | Tree growth, shoot, leaf, and root biomass | Cannavo et al., (2018) |
| | Water storage capacity | Available water content, infiltration | Rossignol (1999); Cannavo et al., (2018) |
| Stormwater management | Microbial diversity and activity | Composition of microbial communities, microbial biomass carbon and nitrogen content, potential net nitrogen mineralization and nitrification, microbial respiration, and denitrification potential, pH, salts | Joyner et al., (2019); Deeb et al., (2018) |





| | | | |
|---|---|---|---|
| | Remove contaminants | Total petroleum hydrocarbons, Pb, Zn, moisture content, texture | Deeb et al., (2018) |
| | Carbon storage | Carbon-nitrogen ratio | Deeb et al., (2018) |
| Urban farming | Biomass production | Edible biomass production (quality and quantity) | Egendorf et al., (2018); Grard et al., (2018) |
| | Carbon storage | Total organic carbon | Grard et al., (2018) |
| | Regulation of water runoff and quality | Water retention, bulk density, particle size density, texture, $NH_4$, $NO_3$ concentration in water, heavy metals concentration in soils and water | Grard et al., (2018) |
| | Biological diversity | Earthworm activity, root development | Grard et al., (2018) |
| | Trace element cycles | Heavy metals concentration in soils and plants | Egendorf et al., (2018); Grard et al., (2018) |
| | Nutrient cycle | pH, available phosphorus, total potassium, extractable manganese and iron, total conductivity | Grard et al., (2018) |
| Reclaim derelict land | Water buffering and transformation | Water retention characteristic, infiltration, hydraulic conductivity, rainfall measurement, drainage effluent | Séré et al., (2008); Smagin (2012) |
| | Biomass production | Organic carbon, total nitrogen, available phosphor, vegetation development and species diversity | Séré et al., (2008); Slukovskaya et al., (2019) |
| | Trace element cycles | Extractable methanol, Cd, Cu, Pb, Zn | Séré et al., (2008) |
| | Soil structure arrangement | Selected descriptors of pores and aggregates | Jangorzo et al., (2018) |
| | Biological diversity | Microbial activity related to the nitrogen cycle, macrofauna activity related to nutrient cycling | Hafeez et al., (2012); Pey et al., (2014) |



930

**Table 2: List of wastes previously used for constructed Technosols in literature.**

| | | |
|---|---|---|
| Mineral | Backfill | Vergnes et al., (2017) |
| | Railway ballast | Yilmaz et al., (2018) |
| | Bricks | Rokia et al., (2014); Vidal-Beaudet et al., (2016); Yilmaz et al., (2018) |
| | Concrete waste | Yilmaz et al., (2018); Pruvost et al., (2018); |
| | Chalcedony | Yilmaz et al., (2018) |
| | Demolition rubble | Yilmaz et al., (2018) |
| | Excavated subsoil | Egendorf et al., (2018); Vidal-Beaudet et al., (2016) |
| | Dredged sediment | Fourvel et al., (2018) |
| | Thermally treated industrial soil | Jangorzo et al., (2013); Séré et al., (2010) |
| | Recycled ferrihydrites | Flores-Ramírez et al., (2018) |
| | Glasses | NYCGIP, (2011) |
| Organic | Green wastes | Yilmaz et al., (2018) |
| | Green waste compost | Grard et al., (2015); Egendorf et al., (2018); Pruvost et al., (2018); Séré et al., (2010); Jangorzo et al., (2018); Grosbellet et al. (2011); Vidal-Beaudet et al. (2012); and Cannavo et al. (2014) |
| | Crushed Wood from public spaces | Grard et al., (2015) |
| | Coffee grounds | Grard et al., (2015) |
| | Sewage sludge compost | Vidal-Beaudet et al., (2016); Yilmaz et al., (2018) |
| | Papermill sludge | Jangorzo et al., (2015); Séré et al., (2010) |





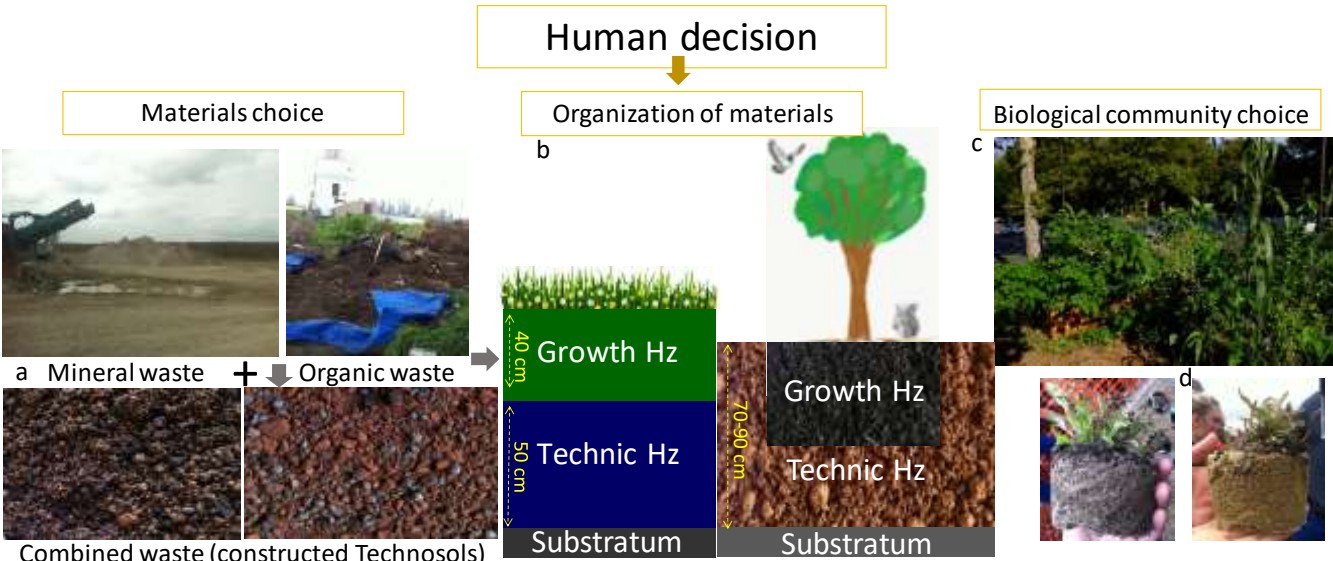

**Figure 1: Diagram presents abiotic and biotic components of constructed Technosols. a) Example of waste mixture during Maha**
935 **Deeb's master's thesis, b) Mixture organized in different layers depending on the land use (SITERRE project) c) Constructed**
**Technosols for urban farming (carbon sponge project). d) Constructed Technosols build by Youth Club for Research and**
**Development in the High School of Landscape and Environment (Vaujours-93).**





Maintenance of green spaces

Soil functions

Key parameters

**Squares and parks**

Lawn mower
Watering
Tree trimming

Support function
Biomass production
Water supply function
and flood regulation
Carbon sequestration

Aggregates stability
Texture
Macroporosity
Available water
N total
P available

**Accompaniment of public building**

Tree trimming
Adding compost and mulch

Water supply function
and flood regulation
Water purification
and contaminant reduction

Carbon sequestration

Aggregation
Texture
Macro-microporosity
Biological function groups
pH
Ksat
Bulk density
Soil depth

**Accompaniment of traffic lanes**

Tree trimming only for safety

Water purification
and contaminant reduction
Carbon sequestration

Biological function groups
pH
Porosity
Bearing capacity

**Urban farming**

Watering
Adding organic material
Harvesting

Biomass and food production
Carbon sequestration

CEC
SOC
N total
P available
K
Porosity
pH
Contaminant-free

**Abandoned land**

No maintenance

Water purification
Habitat for biodiversity
Carbon sequestration

**Figure 2: Key fertility characteristics of constructed Technosols to be considered.**



940

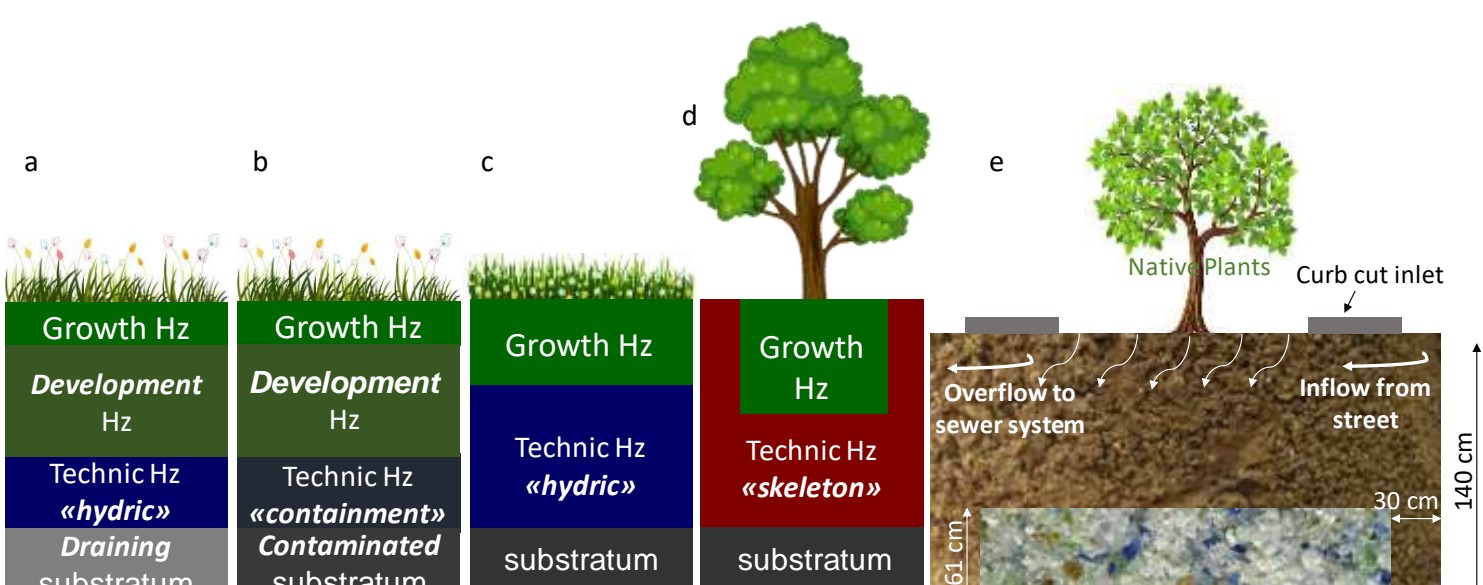

**Figure 3: Examples of the organization of horizons in constructed Technosols noted in literature for: a &b) extensive grassland (Biotechnosol project), c) square and park, d) tree lined streets (c and d from SITTER project), e) stormwater management system (DEPNY, 2010).**



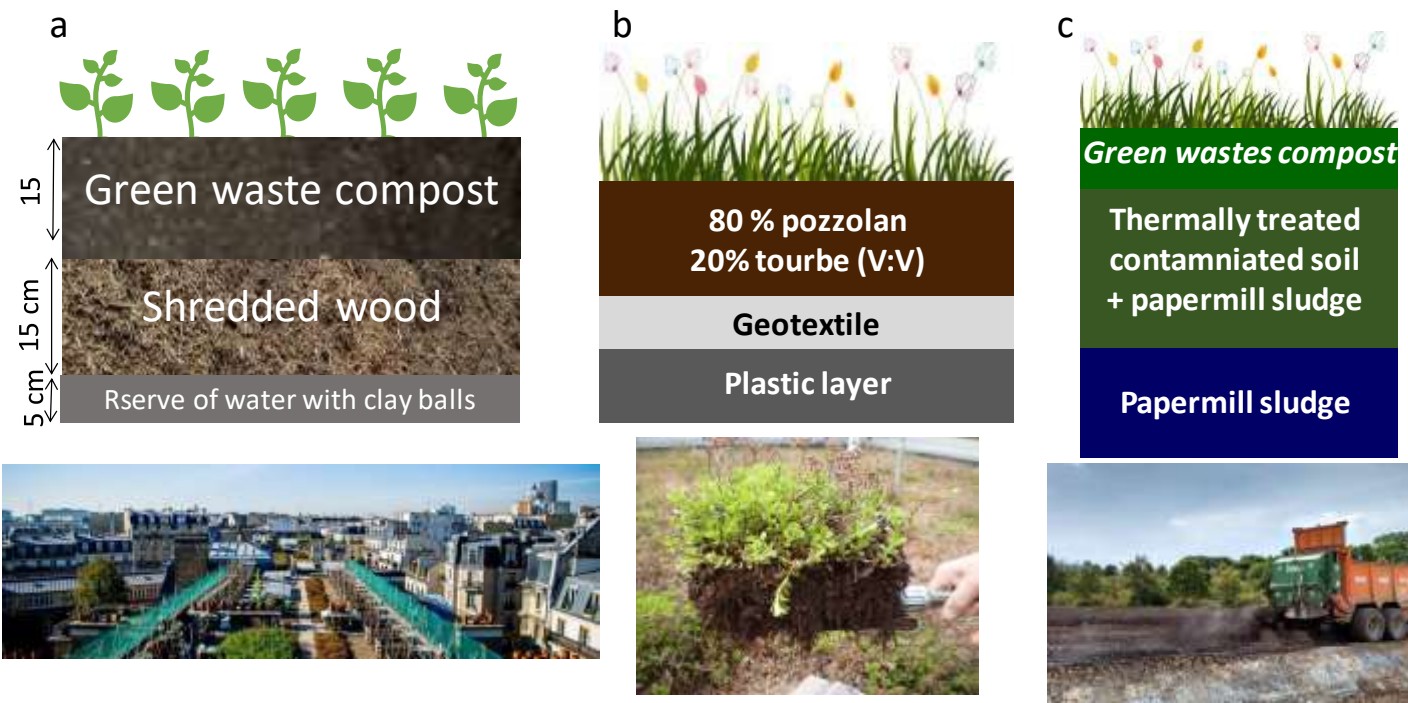

**Figure 4: Examples of the organization of horizons in constructed Technosols noted in literature for: a) green roof (Gerard et al., 945 2018), b) green roof (Bouzouidja et al, 2018), c) abandoned land (Séré et al., 2010).**