# Peer review of "Using constructed soils for green infrastructure - challenges and limitations"

_SOIL, 2019_

## Referee Comment (RC1) · Anonymous Referee #1 · 9 Mar 2020

Dear authors

the article could become very interesting but urgently needs extensions. When you are writing a review article on constructed soils, one would expect an overview of results/findings/suggestions for your suggested categories: 1. Constructed Technosols for parks and squares with lawns 2. Constructed Technosols for developing tree-lined streets 3. Constructed Technosols for stormwater management 4. Constructed Technosols for urban farming 5. Constructed Technosols as a solution to reclaim derelict land I couldn't detect any table or figure describing which ranges of soil properties (chemical, physical and biological) occur or should have and why. The author group is large and experienced enough to distribute these five substrates to design such tables and figures. In these tables/figures, I suggest including international limits i.e. environmental acts in order to show where limitations in the use of urban substrates are. Some more questions raised up: 1. Why these groups, what is the reason i.e. background to distinguish five substrates, where are overlappings? (1 and 2?) 2. Describe and summarize the scientific progress made during the last 10 years 3. Describe the way from science to practice: which substrates are well accepted and used which not and why In addition, I suggest giving a statement on the two questions: 1. are environmental acts limiting factors to produce and use technogenic substrates? 2. what's about plastic and the acceptance of these substrates in the last years? Finally, I suggest modifying the title to 'Using constructed soils for green infrastructure - challenges and limitations'

Your work could be very helpful to describe the next research steps and strategies to accept these substrates. Good luck!

---

## Referee Comment (RC2) · Anonymous Referee #2 · 23 Apr 2020

The authors present a review on constructed Technosols to fulfill specific ecological functions in urban environments. This is a trending topic and is of international interest, since many cities need to recycle different kinds of waste materials in order to diminish landfill area and to avoid using natural soils to construct soils on disturbed terrains. Urban planners are also demanding much more specific advice than currently available in the literature, on which wastes can be used, in which mixtures, how thick the different constructed horizons should be, according to which land use, etc. In this sense I consider that the manuscript could well fill this gap and provide a good overview of the existing experiences.

The authors present an overview of the kinds of waste materials, which have been used so far to construct Technosols, and also show in a first Figure a general diagram of the

steps of the construction process, and another Figure with five different examples, on how substrates can be organized to construct a Technosol depending of the substrate characteristics and the aimed functions. Yet, another Figure (2), with "Key fertility characteristics of constructed Technosols to be considered", is misleading and should be improved. The selection of urban land uses should coincide with the ones stated in Table 1, and the column of "Maintenance" should be moved to the right end, if not deleted (I do not understand what the purpose is of including this information...). The Tables are adequate, however I missed a third Table or Figure that integrates specific characteristics of the different waste materials, which help to achieve the key parameters needed for the different functions. It would be also useful to pinpoint limitations, as for instance ranges of concentrations of heavy metals or other pollutants of the different waste materials, which might counteract the achievement of the desired properties. The review could also include which were the mean physical, chemical and biological properties of the constructed Technosols described in the reviewed papers. Such a list could provide a reference for urban planners, on which values they should be expecting for the mentioned key parameters, when they use different waste materials. A question which was not sufficiently discussed is the proportions in which the different materials should be mixed, or the optimum thickness of the different layers, according to the different land uses. In general I missed more quantitative information or guidance, which I think the review of the different manuscript allows to do.

---

## Author Comment (AC1) · 11 Jun 2020

We thank the reviewers for their comments on the manuscript. We have addressed all of their comments as described below, and feel that the manuscript is greatly improved as a result. Below, the reviewer's comments and our responses follow directly below each comment:

Reviewer #1, "When you are writing a review article on constructed soils, one would expect an overview of results/findings/suggestions for your suggested categories: 1. Constructed Technosols for parks and squares with lawns 2. Constructed Technosols for developing tree-lined streets 3. Constructed Technosols for stormwater manage-ment 4. Constructed Technosols for urban farming 5. Constructed Technosols as a

solution to reclaim derelict land I couldn't detect any table or figure describing which ranges of soil properties (chemical, physical and biological) occur or should have and why."

The reviewer brings up an important expectation that we have long discussed but have been unable to resolve. Below we describe why the search for specific "recipes" for constructed Technosols is difficult and highlight relevant information that is present or has been added to our paper: • As is noted in the paper, the only mathematical model for chemical-physical soil fertility for Technosols was developed in France (Rokia et al., 2014) and was based on comparison of soil fertility characteristics with natural soils. This example provides the information (soil physical and chemical characteristics) the reviewer asked for, but it only addresses one process and does not integrate different contexts, other types of waste, or biological fertility. • In section 3.1, we provide guidance for the choice of substrates based on given criteria and limitations that may occur. However, even here, variance in international norms and access to different substrates poses a great challenge to developing a widely applicable formula. Moreover, one of the objectives of this paper is to encourage readers to adapt to available waste types and by-products, their country's regulations, native biotic factors, desired land use functions, and to highlight the importance of design planning. • Even with one or two specific types of local waste, different formulas are needed for different land uses, For example mixing two types of waste, such as excavated deep horizons with compost, can have several applications for different land uses. While a mix of 30% compost with 70% excavated deep horizon to a depth of 30 cm would be applicable for parks and urban farming, this mixture could also be applied to much deeper depths (1 – 2 m) for tree-lined streets, or below a 5 cm layer of organic mulch to avoid evaporation and conserve water in dry areas. More generally, given the same waste materials, the ratio of compost used will vary greatly depending on the quantity of water received from precipitation. While 20% compost is suitable for open parks in temperate areas, 40-50% compost is needed in dry areas.

Why these groups, what is the reason i.e. background to distinguish five substrates, where are overlappings? (1 and 2?)

We have added the following paragraph (line 160) to the "Methods" section to justify our choices of land uses. "Five land use were chosen for this review based on the land use classification and evaluation provided by Panduro et al.(2013) that included 8 land uses: parks, common area apartments, common area houses, sports fields, agriculture fields, green buffers, nature, lakes. We merged the first three land uses into a single group: "parks and squares with lawns" assuming that these would create similar conditions for the use of constructed Technosols. Sports fields were not included as they have already been discussed in literature (Puhalla et al., 1999). Nature and lakes were excluded as the application of constructed Technosols is not needed in these land uses. Degraded land and tree lined streets were added as complementary independent categories because they are commonly present in urban areas." Consequently, the title of section 3.2.3 has been changed to "Constructed Technosols: Green buffers for stormwater management.

Describe and summarize the scientific progress made during the last 10 years.

Text describing scientific progress made during the last 10 years was added to the Conclusion (before line 435): "Constructed Technosols can contribute to sustainable environments in urban contexts as they supply multiple functions and services in several land uses. Over the past ten years, studies have confirmed the value of mixes that included organic material for soil fertility. A dominant theme that has emerged over this time is mixing excavated deep horizons with organic waste due to the constant need to recycle and repurpose excavated deep horizon waste. Mixtures containing small ratio of natural soils have also been shown to increase the colonization rate of macrofauna. A dominant conclusion that has emerged is that coupling the choice of waste mixture ratios and plants leads to a greater positive impact on soil functions than the choice of waste mixtures alone."

Describe the way from science to practice: which substrates are well accepted and used which not and why In addition,

We have added some text about acceptable substrates to the Conclusion section. "Over the past ten years, studies have confirmed the value of mixes that included organic material for soil fertility. A dominant theme that has emerged over this time is mixing excavated deep horizons with organic waste due to the constant need to recycle and repurpose excavated deep horizon waste". We note that no substrates were rejected based on the conditions listed in section 3.1.2 (Use of waste materials in constructed Technosols). Even low ratios of substrates (i.e., 10% compost with 90% excavated deep horizons) could be used as a B horizon.

I suggest giving a statement on the two questions: 1. are environmental acts limiting factors to produce and use technogenic substrates? 2. what's about plastic and the acceptance of these substrates in the last years?

We have added the following text on "environmental conditions" to the Conclusions section (line 440): "However, this study confirmed that each element used in the formula to design constructed Technosols should be carefully considered. These elements include the ratio and the composition of waste, the order of horizons, environmental conditions, the choice of plant species, the implementation methods, and the critical need to foster pedogenic processes, especially during the first months following construction." There is already a "limitation" section addressing environmental conditions; 3.1.4 - Technical constraints to consider while constructing Technosols. We note that the topic of legislation is briefly mentioned in several places in the text, mostly from the EU and USA perspective. This hasn't been a limitation but rather a motivation for the development and use of Constructed Technosols. We have added some text about plastics to section 3.1.4. - Technical constraints to consider while constructing Technosols (line 241): "Microplastics are another source of contamination that should be considered when building Technosols. Although the number of current studies is limited, plastic contamination may negatively affect plant growth, soil organisms, and

human health through integration in the food chain (Horton et al., 2017). Studies show the use of sewage sludge compost as fertilizer increases microplastic contamination in soils (Zhang and Liu, 2018). To prevent microplastic contamination, sewage sludge compost should be avoided in high quantities, only be used in low ratios, and should be tested for microplastic contamination before application."

"Finally, I suggest modifying the title to 'Using constructed soils for green infrastructure - challenges and limitations'" The title has been changed as suggested.

---

## Author Comment (AC2) · 11 Jun 2020

We thank the reviewers for their comments on the manuscript. We have addressed all of their comments as described below, and feel that the manuscript is greatly improved as a result. Below, the reviewer's comments and our responses follow directly below each comment:

" Figure (2), with "Key fertility characteristics of constructed Technosols to be considered", is misleading and should be improved."

• We have changed "square parks" to "parks and square lawns" and the "accompaniment of public buildings and traffic lanes" to "tree-lined streets". Moreover, instead of stormwater management, we decided to use the term "green buffers" in Table 1 and

[Figure]

Figure 2. ć The "Maintenance" column has been deleted from Figure 2. ć A column for main fertility characteristics of the different wastes has been added to Table 2.

"It would be also useful to pinpoint limitations, as for instance ranges of concentrations of heavy metals or other pollutants of the different waste materials, which might counteract the achievement of the desired properties."

More detail has been added to section 3.1.4. - Technical constraints to consider while constructing Technosols (line 241):

"Fresh organic waste can be problematic as it can have a toxic effect on plant growth (Yilmaz et al., 2016), sometimes by creating anoxic conditions. Thoughtful choices, e.g., using mature compost or mixing in mineral material that drains well (such as sand) can avoid anoxic conditions. Even with these choices, the addition of organic material must be specific to its intended land use as amending organic matter on a regular basis may lead to the accumulation of heavy metals over time. To mitigate this limitation, additional organic matter should be avoided over time to improve Technosol quality and maintain the integrity of established soils. If organic waste must be added, organic matter with heavy metal contamination and pollutants may be mixed with other nontoxic waste ingredients in calculated proportions to lower the overall concentration into acceptable ranges. These ranges will vary according to the land use and local regulations. For example, Total Petroleum Hydrocarbons should not exceed 5000 mg kg-1 as defined in the European Union (Pinedo et al., 2013). In New York City, heavy metals and organic contamination limits strictly depend on land use NY-CRR (2017)."

We did not list specific ranges for heavy metals and other pollutant concentrations because these ranges vary greatly depending on the limitations imposed by individual countries and even states. In the review, we make the case that materials should be clean to begin with. We also provided a citation that lists the concentration ranges of heavy metal and organic contamination limits for different land uses in New York City.

"The review could also include which were the mean physical, chemical and biological properties of the constructed Technosols described in the reviewed papers. Such a list could provide a reference for urban planners, on which values they should be expecting for the mentioned key parameters, when they use different waste materials. A question which was not sufficiently discussed is the proportions in which the different materials should be mixed, or the optimum thickness of the different layers, according to the different land uses. In general I missed more quantitative information or guidance, which I think the review of the different manuscript allows to do."

• We note that several specific examples of thickness and mixture ratios from literature are provided for each land use. However, as noted above, there is no universal formula for Constructed Technosols due to the complexities of local waste material, biotic choices (plants, macrofauna), and climate conditions. • We also note that the current literature available is not extensive and does not allow us to draw general conclusions about waste ratio and thickness to provide a recipe that everyone can use.

———————————————————

[Figure]

[Figure]

| Soil functions | Key parameters |
|---|---|

**Parks and squares lawn**
Support function
Biomass production
Water supply function
and flood regulation
Carbon sequestration
→
Aggregates stability
Texture
Macroporosity
Available water
N total
P available

**Tree-lined streets**
Water supply function
and flood regulation
Water purification
and contaminant reduction
Carbon sequestration
→
Aggregation
Texture
Macro-microporosity
Biological function groups
pH
Ksat
Bulk density
Soil depth

**Green buffers**
Water purification
and contaminant reduction
Carbon sequestration
→
Biological function groups
pH
Porosity
Bearing capacity

**Urban farming**
Biomass and food production
Carbon sequestration
→
CEC
SOC
N total
P available
K
Porosity
pH
Contaminant-free

**Reclaim derelict land**
Water purification
Habitat for biodiversity
Carbon sequestration

**Fig. 1.**

---

## Editor Decision (ED1)

Final review and topical editor´s comments

The authors considered reviewers' comments and adjusted the text quite accordingly.

Still, I like to suggest some minor adjustments, mostly related to structure and text:

- Figure 2 needs to be improved in captions and a few terms. My suggestion is: Soil properties to be considered to fulfill soil functions provided by constructed Technosols in different land uses.

Key parameters inside the figure should be Soil properties.

In both cases, I would not use Key, since they are many and there is not enough indication that some are keys and others not…. This also happens in other parts of the text as in line 170, where Characteristics is more realistic than Key characteristics. In table 1, also I suggest to replace key parameters by Soil characteristics of CT considered to assess soil functions by different land uses of GI, provided by literature.

The term fertility is used here and also in the text to refer to different soil properties that are not only chemical, but sometimes physical, and even more important for the purposes and land uses. This should be adjusted throughout the text, as for instance in line 455, in the Conclusions

I also suggest to use Reclaimed derelict land instead of reclamation to keep the same text pattern.

- In the conclusion, line 464, the expression "in the formula" may not apply. I suggest to remove it.
- In the structure of the results, I suggest some text adjustmens/arrangements to improve the text pattern:

3.1 Construction of Technosols for different land uses in GI

3.1.1 Needed characteristics to fulfill GI requirements

3.1.2 Use of waste materials

The item 3.1.3 is not relevant for itself, being part of the previous one, as another paragraph: To be functional, a specific…. I suggest to remove it as a separate item

3.1.3 Technical constraints (former 3.1.4)

3.1.4 Pedogenesis

3.2 Construction of Technosols to specific land uses in GI (I suggest this the keep an uniform terminology that eases reading) followed by subtitles harmonized with the ones in Figure 2

3.2.1 Parks… 2 Tree lined… 3 Green buffers 4 Urban farming 5 Reclaimed derelict land

With those minor adjustments I consider that the paper is ready to be published.

I like to thank the reviewers for their comments and suggestions and wish success to the authors.

---

## Author Response (AR2)

Dear Editor,

Please find attached a revised manuscript (Soil-2019-85) to be considered for publication as a review in a Soil journal. We thank the Editor for their interest and their comments on the manuscript. We have addressed all of the comments as described below, which undoubtedly improved the manuscript overall. In addition, we made some grammatical changes throughout the article, tables, and figures as indicated below.

- "Figure 2 needs to be improved in captions and a few terms. My suggestion is: Soil properties to be considered to fulfill soil functions provided by constructed Technosols in different land uses.
  Key parameters inside the figure should be Soil properties. In both cases, I would not use Key, since they are many and there is not enough indication that some are keys and others not…. This also happens in other parts of the text as in line 170, where Characteristics is more realistic than Key characteristics".

  *This change has been made.*

- In table 1, also I suggest to replace key parameters by Soil characteristics of CT considered to assess soil functions by different land uses of GI, provided by literature.

  *Done.*

- "The term fertility is used here and also in the text to refer to different soil properties that are not only chemical, but sometimes physical, and even more important for the purposes and land uses. This should be adjusted throughout the text, as for instance in line 455, in the Conclusions I also suggest to use Reclaimed derelict land instead of reclamation to keep the same text pattern.

  *In Table 2, "Fertility" was changed to "Main physical / chemical characteristics".*
  *In line 242, "fertility" was changed to "biomass production".*
  *In line 417, "fertility" was changed to "physico-chemical characteristics".*
  *In line 455, "fertility" was changed to "soil functions".*

- In the conclusion, line 464, the expression "in the formula" may not apply. I suggest to remove it.

  *Done.*

- In the structure of the results, I suggest some text adjustments/arrangements to improve the text pattern:

  3.1 Construction of Technosols for different land uses in GI
  3.1.1 Needed characteristics to fulfill GI requirements
  3.1.2 Use of waste materials The item 3.1.3 is not relevant for itself, being part of the previous one, as another paragraph: To be functional, a specific…. I suggest to remove it as a separate item
  3.1.3 Technical constraints (former 3.1.4)
  3.1.4 Pedogenesis

3.2 Construction of Technosols to specific land uses in GI (I suggest this the keep an uniform terminology that eases reading) followed by subtitles harmonized with the ones in Figure 2

3.2.1 Parks… 2 Tree lined… 3 Green buffers 4 Urban farming 5 Reclaimed derelict land

*The text was modified as you suggested.*

In addition to your suggestions, some grammatical errors in the manuscript were corrected, as follows:

**Abstract:**

In line 27, we changed the list to reflect the new modifications made in the article: "... constraints to using CT for applications in parks and square lawns, tree-lined streets, green buffer for stormwater management, urban farming, and reclaimed derelict land."

**Table:**

In Table 1, the names were updated to be "parks and square lawns", "tree-lined streets", and "reclaimed derelict land".

**Figures:**

-In Figure 2, the names were changed to "parks and square lawns", "tree-lined streets", and "reclaimed derelict land".

-In Figure 3, SITTER was corrected to "SITERRE", square and park was changed to "parks and square lawns", and tree lined streets to "tree-lined streets".

-In Figure 4, the diagram (a) was corrected to "Reserve" and diagram (c) to "contaminated".

[revised manuscript text omitted]